# Integration of electrically detected magnetic resonance on a chip (EDMRoC) with charge pumping for low-cost and sensitive defect characterization in SiC MOSFETs

Jan Lettens<sup>1,2</sup>, Marina Avramenko<sup>2</sup>, Ilias Vandevenne<sup>1</sup>, Anh Chu<sup>3,4</sup>, Philipp Hengel<sup>3,4</sup>, Michal Kern<sup>3,4</sup>, Jens Anders<sup>3,4</sup>, Peter Moens<sup>2</sup>, Etienne Goovaerts<sup>1</sup>, Sofie Cambré<sup>1</sup>

<sup>1</sup>Department of Physics, University of Antwerp, Antwerp, 2610, Belgium
 <sup>2</sup>onsemi, Westerring 15, Oudenaarde, 9700, Belgium
 <sup>3</sup>Institute of Smart Sensors, Universität Stuttgartm Stuttgartm 70569, Germany
 <sup>4</sup>Center for integrated Science and Technology (IQST), Stuttgart, 70569, Germany

Correspondence to: Sofie Cambré (sofie.cambre@uantwerpen.be)

**Abstract.** Integration of microwave sources and detection circuits has led to the design of very compact electron paramagnetic resonance (EPR) instruments, so-called EPR on-a-chip (EPRoC). As recently demonstrated, this approach also offers opportunities for electrical detection of magnetic resonance (EDMR), a variant of EPR in which the magnetic resonance effect is detected via changes in the electrical properties of materials or devices. Here, we report the demonstration

- of EDMRoC on lateral SiC MOSFETs under charge pumping (CP) conditions. The detected CP current gives direct access to microscopic information about the recombination centers within the transistor gate inversion region under the gate dielectric. Efficient and selective microwave excitation of the region of interest of the device can be obtained by only modest modifications to both the MOSFET and the EPRoC electronic board. A comparative study between EDMRoC and a traditional resonant cavity configuration reveals comparable signal-to-noise ratios for CP-detected EDMR spectra. In
- addition to space- and cost-efficiency, EDMRoC offers alternative detection modes with scanning and modulation of the microwave frequency, as well as potentially easier sample mounting and exchange. We end with a discussion of the advantages, limitations, and perspectives of the EDMRoC set-up compared to EDMR in a conventional EPR spectrometer.

### **1** Introduction

Electrical detection of magnetic resonance (EDMR), probing electron paramagnetic resonance (EPR) via changes in electrical properties due to spin-dependent processes, has become a method of choice for microscopic characterization of performance-limiting defects in semiconductor materials and devices.(Boehme and Malissa, 2017) The latter includes 2terminal devices such as diodes and solar cells,(Akhtar et al., 2015; Anders et al., 2018; Boehme and Malissa, 2017; Lepine, 1972; Li et al., 2004; Malissa et al., 2014; Rong et al., 1991; Tedlla et al., 2015) but also more complex devices such as metal-oxide-semiconductor field-effect transistors (MOSFETs).(Ashton et al., 2019; Bittel et al., 2011; Cochrane et al.,

2010; Cottom et al., 2018; Umeda et al., 2011, 2018, 2019) Recombination centers in the transistor channel area are directly

related to performance limitations and degradation during operation in these devices, in particular in the technologically important case of silicon carbide (SiC) MOSFETs.(Allerstam et al., 2007; Bathen et al., 2022; Harada et al., 2002; Wang et al., 2023) Therefore, EDMR spectroscopy has been applied early on for defect characterization using different detection schemes based on changes in transimpedance between different terminals of the transistor. (Aichinger and Lenahan, 2012; Ashton et al., 2019; Cochrane et al., 2010, 2012; Cottom et al., 2018; Kagoyama et al., 2019; Sometani et al., 2023; Umeda 35 et al., 2011, 2018, 2019) However, in our experiments, we apply the spin-dependent charge pumping EDMR scheme (SDCP or CP-EDMR) as first demonstrated by (Bittel et al., 2011), which is based on the well-established charge pumping (CP) method for transistor characterization.(Van den bosch et al., 1991; Brugler and Jespers, 1969; Djezzar, 2023; Groeseneken et al., 1984; Lettens et al., 2023; Okamoto et al., 2008) The CP current monitored in this scheme is induced by a rectification 40 effect between the base and source (and/or drain) terminals under excitation of the gate by a voltage periodically alternating between two voltage levels. These two levels are chosen such that alternatively either electrons or holes are attracted to and extracted from the transistor channel region, and recombination occurs at trapping sites. Albeit more intricate, the CP version of EDMR can offer a direct correlation with the results of CP characterization.(Aichinger et al., 2013; Anders et al., 2020; Bittel et al., 2011; Cochrane et al., 2013; Gruber et al., 2014, 2018) It potentially also offers higher selectivity in several 45 ways: (i) by the choice of applied voltage levels that may be energetically aligned to specific defect levels, and (ii) by variation of other CP-excitation parameters (frequency, rise/fall times, duty cycle) which help to discriminate between traps with different temporal characteristics and/or spatial distribution. Although EDMR has been frequently applied to investigate defects in MOSFETS, the requirement of (expensive) dedicated equipment prevents it from being routinely used. Therefore, a non-resonant EDMR application, replacing the microwave (MW) resonator of the conventional EDMR by a non-resonant

antenna has been successfully applied to measure frequency-swept and rapid scan EDMR spectra.(McCrory et al., 2019)

In recent years, developments in MW electronics have led to so-called EPR-on-a-chip (EPRoC), – a single-board EPR set-up with an integrated MW oscillator chip –, which allows for extremely compact and low-cost EPR instrumentation.(Anders et al., 2012; Yalcin and Boero, 2008) In conventional EPR experiments, the sample is inserted in a high-Q MW resonator providing standing waves with high amplitudes of the magnetic field component and minimal electrical field components at

- providing standing waves with high amplitudes of the magnetic field component and minimal electrical field components at the sample position, for high detection sensitivity and low dissipation losses, respectively (see Fig. 1(d) for a photograph of a typical setup). Often, the insertion of an electronic device in a resonator is not straightforward, considering both the specific requirements in size and geometry and the presence of metallic conductors interfering with the MWs, as discussed in detail in (Segantini et al., 2023). On the contrary, in EPRoC, the MWs are generated using a voltage-controlled oscillator (VCO)
- integrated using commercial silicon chip fabrication technologies. The coils in the VCO serve both for generation of the near-field MW fields as well as for inductive detection of spin signals. Samples are positioned in the proximity field of this MW chip (see comparison in Fig. 5 of (Segantini et al., 2023)). This was shown to be versatile, with a wide range of possible MW frequencies (up to a first harmonic oscillation frequency of 263 GHz,(Chu et al., 2023) even several frequencies combined as well as their higher-order harmonics (Handwerker et al., 2016; Matheoud et al., 2017)) and it allows frequency

- scanning/modulation (replacing conventional static magnetic field scanning/modulation). Moreover, the MW chip, which can be adapted in size and shape, is very well suited for excitation of thin planar samples. This was very recently exploited by (Segantini et al., 2023) in a demonstration of EDMR-on-a-chip (EDMRoC) in a thin film a-Si:H solar cell, using the EPRoC MW source and detecting EDMR via the change in conductivity of the photovoltaic active layer under forward bias.
- Here, we extend EDMRoC spectroscopy to measurements of EDMR on lateral SiC MOSFETs using the powerful CP detection scheme mentioned above. The MOSFET chips are well matched to the dimensions of the EPR chip and mounted in a way appropriate for measurements either in the EDMRoC approach using the EPRoC MW source, or in the resonator of the X-band EPR spectrometer. As a result, direct and quantitative comparisons were obtained, showing a signal to noise ratio (SNR) comparable in EDMRoC with the optimized conventional experiment. Results will be shown with varying
- instrumental parameters and using different scanning/modulation methods. Finally, a compact set-up using a permanent magnet is demonstrated.

# 2. Results and Discussion

# 2.1. Design of EDMRoC for SiC MOSFETs

### 2.1.1. Microwave Generation

- The basis of our EDMRoC experiments is the printed circuit board (PCB) shown in Fig. 1(a) and Fig. 1(b), providing an integrated MW source. It is very similar to the one applied in (Segantini et al., 2023) however without the transimpedance amplifier (TIA). As discussed below, we rather use the external TIA employed for EDMR in our conventional EPR setup because of the specific requirements for detection via CP. On this main board, MWs are generated by an application-specific integrated circuit (ASIC) carrying an injection-locked array of voltage-controlled oscillators (VCO) with variable frequency
- in a range of width  $\approx 2.5$  GHz centered around the design frequency of 14 GHz. The VCO-array oscillation frequency is divided on-chip and made available off-chip by a ratio of 32. This allows for a custom-made phase-locked loop (PLL) on the PCB to lock the divided frequency to the output frequency of a radio frequency (RF) generator (Rhode&Schwarz SMC100A). The EPR chip (see Fig. 1(b)), with two rows of six 200 µm octagonal coils each, placed apart 300 µm center-tocenter within one row and 425 µm center-to-center in between the two rows, generates MWs linearly polarized with the
- magnetic field  $B_1$  perpendicular to the surface, quite rapidly decreasing with distance from the chip surface. The generated MW power can be adjusted and monitored via the summed control current  $I_{BIAS}$  (17.5-300 mA),(Künstner et al., 2021) with a threshold current  $I_{BIAS} \cong 17.5$  mA and a smooth and continuous behavior up till the highest current values  $I_{BIAS} \cong 300$  mA (see Appendix Fig. A1(b)). The quantitative comparison with MW powers supplied in the resonant cavity set-up will be discussed below (see Fig. 3 and Appendix A and B).