# Peer review of "Integration of electrically detected magnetic resonance on a chip (EDMRoC) with charge pumping for low-cost and sensitive defect characterization in SiC MOSFETs"

_Magnetic Resonance, 2024_

## Author Comment (AC1)

**2.1 RC1: 'Comment on mr-2024-11', Anonymous Referee #1, 10 Aug 2024**

We would like to thank the reviewer for their assessment of our manuscript. We have replied to the different comments separately below. Note that we indicate the line numbers as in the new version of the manuscript.

1. *The EPR chip is small, but the entire experimental setup includes many components (synthesizer, current amplifier, etc.). It would be helpful to show all these devices in a schematic drawing and highlight the role of the EPR chip and what it saves over a conventional setup. Specifically, it would be useful to understand if the EPR chip in this context can simply be replaced by a small broadband antenna.*

As mentioned in the manuscript, the EPR chip with external compact magnet replaces the microwave source, the microwave resonator (and its intrinsic restriction to use a fixed microwave frequency) and the sweepable electromagnet. All other components are conventional scientific instruments, that are relatively cheap and can be easily found in many laboratories. Note that it can in principle even be much cheaper than the demonstrated instruments, as many of those we used in this work are multi-purpose lab instruments with different functionalities that are not fully required for this application.

The setup is schematically illustrated in a new figure in the appendix I, Figure I1.

[Figure]

**Figure I1: Schematic setup of the experiment. For the charge pumping we employ a HP33120A function generator, giving a sinusoidal excitation with an amplitude $V_A$ of 10 V. The base voltage VB can be varied using a DC power supply. This sinusoidal excitation is provided to the gate of the device, with the source and drain grounded and the charge pumping current is measured at the body of the device, amplified by the TIA (SR570) to a detectable input voltage for the lock-in amplifier. For cavity-based EDMR, a microwave source is combined with a microwave cavity, in between the poles of a sweepable electromagnet, while for EDMRoC the MWs of the chips are locked on an RF source (SMC100B), off which the 32$^{nd}$ harmonic is used to generate the MWs. The required frequency of ~450 MHz could readily be obtained from a more basic RF source. The $I_{VCO}$ is of the EDMRoC is provided by a standard DC current source. For frequency modulation, the modulation signal is provided to the lock-in amplifier as a reference, while for magnetic-field modulation, both in cavity-**

**based EDMR and EDMRoC, the reference to the lock-in amplifier comes from the control unit of the EPR spectrometer. Note that for all experiments reported here, we made sure that the MOSFET device, charge pumping parameters as well as amplifier and lock-in amplifier settings were all kept identical, so that EDMRoC and cavity-based EDMR can be ideally compared. For the lock-in amplifier, the SPU unit of our Bruker EPR spectrometer is used, but it can be replaced by any lock-in amplifier for EDMRoC.**

Replacing the MW ASIC on the EDMRoC board by an antenna is not possible, since the VCOs are an integrated part of the MW generation circuitry. One could, in principle, replace the EPR chip by a broadband antenna, but at a significant cost in complexity, size of the setup and overall costs. For example, if one would like to achieve comparable excitation volume with comparable $B_1$ strength at the same frequency, one would need a 14 GHz signal generator combined with a power amplifier with significant power output. The advantage of the EPR chip is that it only needs an approximately 450 MHz frequency reference with frequency modulation capability, here provided by the SMC100B. This instrument is however overkill and could in a later design be replaced with a single PCB. Moreover, as we mention in the discussion section of our manuscript, the MW ASIC is scalable (not limited to the 2 rows of 6 coils) and can thus be adapted to match smaller or larger planar samples and thus provide a larger excitation volume.

Importantly, as demonstrated previously, one can combine the EPR chip with a PCB for delivering the reference frequency with modulation capability, and another PCB with the necessary power supplies and lock-in amplifier for signal detection, which could fit the whole spectrometer into 10x10x10 cm$^3$ operated by a battery [doi: 10.1109/ISSCC.2016.7418114]. The prize of such a spectrometer would depend on the number of spectrometers that are produced but amounts to a few hundred euros per chip depending on technology, chip size and demand (in the case of a demand of less than 100 units). For medium-scale production, the cost per chip will decrease to less than hundred euros, making it a low-cost solution comparable to a typical cavity-based EDMR spectrometer that can easily amount up to 500.000 euro.

We added a remark regarding this on lines 395-398 in the revised manuscript:

*Finally, it is important to note, that the reference frequency of the VCO (here provided by the SMC100B), the DC power supply and the current source ($I_{VCO}$) to operate the EDMR chip can in principle be replaced by small dedicated PCBs that can even operate with a battery, so that the entire setup can fit in a very small spectrometer, as previously demonstrated in (Handwerker et al., 2016), further reducing the costs of the system to just a few hundred euros.*

2. ***It would be beneficial to add a figure that summarizes and explains the CP mechanism in the MOSFET device.***

The essentials of CP are described in the introduction of our manuscript and in section 2.1.4 "Charge pumping and EDMR Detection" of the manuscript. We have also presented a schematic drawing of CP in Figure 2(f) and the necessary equipment is added in the new Figure I1. CP is a commonly used technique for characterizing defects in MOSFET devices, and references to the literature are provided (e.g. Groeseneken 1984 and Okamoto 2008 and Lettens 2023) which explain the technique in detail. Since it is not essential to the current manuscript, CP being used to generate a direct current which is known to be spin-dependent, we decided to not complicate the manuscript with a more detailed explanation.

3. ***Regarding the text "MOSFET device," does it resemble a real device, or is this just a test sample? What does a real sample look like, and what are the prospects of measuring a real sample?***

As explained in section 2.1.2 in our manuscript ("The MOSFET device") as well as in section 2.13 ('Mounting of the MOSFET device"), it is a lateral n-type MOSFET test device. Commercial SiC power MOSFETs are vertical transistors, with the drain contact at the bottom of the die (SiC substrate). These

devices always have shorted source/body contacts, which makes the measurement and interpretation of charge pumping more challenging. Typically, for advanced characterization, simple test-structures are used, in our case a lateral n-type MOSFET. These devices are much simpler and also have a separate source/body contact which allows to measure the hole recombination current which is required in charge pumping. The source and $p$-body, gate oxide and gate poly and metal are the same as for the vertical SiC power MOSFETs. Hence the results extracted on a lateral test-structure about the SiC/SiO$_2$ interface are directly transferable to a vertical device.

Recently, CP EDMR has been reported on such commercial SiC power MOSFETs [Lew et al., J. Appl. Phys. 134, 055703 (2023); doi: 10.1063/5.0167650], hence there is in principle no limitation to apply CP-EDMRoC also to commercial power MOSFETs, and in this case the EDMR chip can be adapted to match the size and excitation volume of those MOSFETs.

We have added the following sentences in section 2.1.2 of our revised manuscript (lines 124-128)

*"Note that commercial SiC power MOSFETs are vertical transistors, with the drain contact at the bottom of the SiC substrate. These devices always have shorted source/body contacts, which makes measurements and interpretation of charge pumping more challenging. Therefore, lateral test structures are devised with identical materials and processing of the source, drain and body, the gate and gate oxide, as in the vertical SiC MOSFETs, so results are directly transferable."*

We have also added the following sentence in section 3 of our manuscript and added a reference to this paper (lines 351-352 in revised manuscript):

*"While in this work we focused on lateral test MOSFETs, a recent report demonstrated the power of CP-EDMR also on commercial SiC MOSFETs, showing the versatility of CP-EDMR.(Lew et al., 2023)"*

**4. Figure 2c - Where are the n, p, and gate parts in the structure shown in 2c?**

Figure 2(c) is a top view picture of the MOSFET device showing contrast based on the surface materials, in particular metallization and a-SiO2 isolator, hence the n, p and gate parts are difficult to indicate. To clarify, we have added the Source (S), Drain (D), Body (B) and Gate (G) contact pads in Figure 2(c). This can then be directly related to the schematic cross-section of the device presented in figure 2(f) where the different contacts are given. The body is the p+ contact where $I_B$ is measured, while source and drain are the n+ contacts. The gate is in black.

We have thus adapted Figure 2 accordingly and added this information to the caption of Figure 2.

**5. Figure 2e - Please add a scale bar to this figure to appreciate the distances.**

We have added a scale bar to this panel.

**6. Line 157 - An illustrative figure for the spin resonance effect in the measured material would be helpful to include.**

In the (quite extensive work) on this topic in literature only fragmentary explanation of the spin resonance effect for EDMR in SiC MOSFETs has been previously provided, and this is even more the case for CP-EDMR. We have cited the most relevant literature in the introduction of our manuscript (pages 1-2). This manuscript reports on an alternative experimental approach for these measurements, independent on which spin-dependent current is probed. We estimate that the more detailed description of the spin resonance effect is beyond the scope of this report.

7. *Line 205 - It would be useful to see the B1 field superimposed on the measured sample in Figure 2c.*

We estimated that adding the calculated $B_1$ field superimposed on the sample in Figure 2c would make the figure overcrowded. Therefore, we had previously added an overlay of Figure 2c on top of the microwave chip in appendix D (Figure D1). Moreover, in appendix G1, the $B_1$ field is presented in figure G1, hence we believe this was already provided, be it in two complementary figures in the appendices of the manuscript. We have therefore added the following sentence on line 232 of the revised manuscript:

*Note the overlap of our MOSFET sample with the microwave ASIC, as presented in Appendix D, Figure D1.*

8. *Line 257 - This section is repetitive of the previous subsection. Consider rearranging it to avoid redundancy.*

While in the previous section 2.2.1 we explain how to rescale the different MW powers in the two setups, in section 2.2.2 we make the actual comparison between both systems. We therefore only found one sentence overlapping, which we have deleted:

**

To delete this sentence, we added on line 276: "*..... at matching effective power levels.*"

9. *What is the (calculated?) static magnetic field profile/homogeneity for the permanent magnet?*

We do not have a calculation of this field profile. In the current simple design, the positioning of the sample is thus critical (within 0.5mm accuracy) to obtain optimal line widths. Moving the sample in any direction of this optimized position results in broadening of the line shape. Since permanent magnets with better field inhomogeneity are for sale at reasonable prices, it would only require a minor investment to improve this field homogeneity.

We did already mention this briefly in the manuscript on lines 332-334 and mentioned that this is the current limitation of the simple design, but have now further added the following sentences to stress this:

*Note that the small line broadening observed for the permanent magnet assembly most likely originates* from a ** less **homogeneous $B_0$ *field in this very simple design.* **Indeed, the exact sample positioning needs to be accurate (within 0.5 mm) to obtain this specific line width and position of the EDMR spectrum.** *While this demonstrator evidently calls for further smart design of the magnet/sample set-up (e.g., with better field homogeneity and/or easier access to the sample area) and of other components of the spectrometer, it gives a taste of the possibilities opening up for ultra-compact, low-budget and dedicated EPRoC and EDMRoC spectrometers.*

10. *Line 360 - What about the use of methods like those in 10.1016/j.jmr.2015.02.010?*

In the latter very interesting paper (Hubresch et al., JMR2015) the focus is on applications of pulsed magnetic resonance using sophisticated generation techniques for the microwaves, including arbitrary wave generation (AWG) combined with up- (and down-) conversion which is justified to reach the aimed spectroscopic resolving power. It brings requirements on maximum size of the sample for sufficient $B_1$-field homogeneity as well. One also has to note that the excitation volume in this approach is much smaller than in our case, and due to the small diameter of the excitation short-circuit, one needs to deposit the excitation electrodes directly onto the sample of interest, as the $B_1$ decays with distance even more drastically than in our case.

Our approach is quite different and aims at a robust, low-cost spectrometer for sensitive detection of continuous-wave EDMR in operational MOSFET devices. We have demonstrated that the use of the MW generation based on VCOs efficiently serves this purpose. At this point, there is no straightforward path to applications of this platform for pulsed EDMR as in the proposed manuscript. Importantly, as mentioned in our answer to this referee's first remark, the EPR chip can theoretically be operated by a portable battery and reach $B_1$ strengths comparable to conventional setups.

---

## Author Comment (AC2)

We thank the reviewer for the careful consideration of our manuscript and the appreciation for our work. We would like to point out that since this is a technical paper on a significant advancement of the use of EPRoC in the field of EDMR, we found it highly suitable for this journal which focuses on advances in all fields of magnetic resonance, also in the techniques supporting the experiments, which is exactly the focus of our manuscript.

General comments:

1. *There is a large number of publications using EDMR applied to solid state device, among others transistors, Schottky and pn diodes and solar cells that date back to the last century. I recommend to also cite that work, specifically the work of Martin Brandt and collaborators (e.g. Genshiro Kawachi et al 1997 J. Appl. Phys. 36 121), who have also performed fundamental work developing the EDMR technique.*

The reviewer is indeed correct that conventional cavity-based EDMR has been applied to many types of solid-state devices and therefore it was impossible to cite all related manuscripts in this work. We do agree that the early work of Martin Brandt is important in this perspective and added in the introduction of our manuscript (line 29) two citations: Kawachi et al., 1996, Phys. Rev. B 54, 7957; https://doi.org/10.1103/PhysRevB.54.7957 as well as the paper mentioned by the referee.

2. *The readability of the paper would be dramatically enhanced if a scheme of the electronic processes of the MOSFET are included. The authors should show and discuss the specific spin-dependent transitions and their influence on the recombination current and the functionality of the device.*

As we also replied to reviewer 1, the aim of our article is the comparison of EDMRoC with cavity-based EDMR, with CP as the source of the spin-dependent current as amply documented in cited work by Lenahan and collaborators (reference Bittel et al, 2011; and later work) and other research groups. CP is a commonly used technique for characterizing defects in MOSFET devices, and references to the literature are provided (e.g. Groeseneken 1984 and Okamoto 2008 and Lettens 2023) which explain the technique in detail. Moreover, the essentials of CP are described in the introduction of our manuscript and in section 2.1.4 "Charge pumping and EDMR Detection" of the manuscript. We have also presented a schematic drawing of CP in Figure 2(f) together with a description of the experiment, the used parameters, and the resulting CP curve as a function of base voltage in Figure 3(a), giving the necessary background information. Additionally, we have included Figure I1 showing the entire setup.

We consider that the exact mechanisms of the spin-dependent transitions and its influence on the recombination current lies beyond the scope of our manuscript and decided to not complicate the manuscript with a more detailed explanation.

3. *The paper of McCrory (ref. McCrory 2019) demonstrates how the limitation for the EDMR application is indeed the resonator and not the price of the instruments. McCrory et al. also show many new applications and experiments that can be performed using a different detection scheme. The authors should comment on this.*

We have indeed cited the work of McCrory et al., 2019, as a very interesting demonstration of EDMR avoiding the constraints of the microwave resonator. There are however large differences in purpose and concepts with our work. They focus on integration of the EDMR characterization in the semiconductor wafer-probing station while EDMRoC offers the perspective of a tabletop low-cost instrument with

sensitivity comparable to the best cavity-based EDMR setup. It is not straightforward to imagine a combination of the latter with a probing station, but part of the constraints of device dimensions and the need for mounting and bonding onto a sample PCB were now imposed by the need for parallel measurements in the MW resonator. In further realizations, the size of the MW emitting chip is adjustable to needs, and other electrical contacting approaches of the device can be considered.

In the very interesting work of McCrory et al., the essential goal is to combine EDMR with wafer level testing and evaluation of the MOSFETs, for which the small and agile antenna probe at the end of a coaxial cable is well-chosen. While this is optimal for the use in wafer-level testing, the dependence on positioning and orientation must be critical for the excitation efficiency. Also in our EDMRoC approach there is a need for more accurate control on the sample to MW chip distance (see lines 354-355 in revised manuscript), but this can be solved by improved mechanical mounting.

Also, such an antenna is in general sub-optimal from the point of view of magnetic resonance as it maximizes electromagnetic radiation (both $E_1$ and $B_1$ fields) while the coils in the MW ASIC are optimized for maximum $B_1$ and minimal $E_1$ emission. The latter is contributing positively to the very low heat dissipation by MW absorption we demonstrated in EDMRoC (subsection 2.2.1; lines 263 - 265 of revised manuscript).

In their article, McCrory *et al.* do not give any technical information on the MW source and amplifier needed to obtain similar local power in the proximity of the antenna as one finds using a high Q resonator, making it hard to compare the required peripheral instrumentation and their cost. As described in detail, the PCB used in EDMRoC is self-contained for MW generation, needing only a DC power supply and a relatively low-frequency reference signal.

Comparing the versatility of both approaches, their antenna is broadband allowing for a wide frequency range (of course provided sufficiently powerful MW sources are available), compared to a limited scanning range in this EDMRoC implementation (however for EPRoC realizations at different and even several simultaneous frequencies have been demonstrated: Anders et al, 2012; Handwerker et al, 2016; Matheoud et al, 2017; Chu et al., 2023). Frequency scanning and modulation are common features in both these approaches. As to the rapid-scan technique: this would in principle also be possible in EDMRoC following the path of rapid-scan EPRoC (in Künstner et al, 2021). There seems to be no fundamental obstacle to apply EDMRoC in other situations, including other devices (diodes, solar cells, …) and other detection modes for MOSFETs (diode current; bipolar amplification effect, BAE). However, here we choose for a detailed comparison with resonator-based EDMR in the case of the relatively demanding CP-based EDMR, and so we could provide solid evidence that the compact EDMRoC setup is versatile and offers competitive detection sensitivity. The EDMRoC approach opens broader perspectives which however still need to be further explored and evaluated.

We have explained the differences in a better way on line 52 of the manuscript by adding the following:

*Therefore, a non-resonant EDMR application, replacing the microwave (MW) resonator of the conventional EDMR by a non-resonant antenna has been successfully applied to measure frequency-swept and rapid scan EDMR spectra, **allowing for direct measurements of EDMR in a wafer-scale probing station.**(McCrory et al., 2019)*

   **4. *A reference should be given that describes in a bit more detail how the MOSFETs have been prepared.***

In section 2.1.2 the MOSFET production is fully described. These are standard lateral test structures.

5. *A definition of the EDMR signal intensity should be given. Usually the MW-induced current change is normalized to the total current. This, however, seems not to be the case here. The authors should explain, why they choose to use ΔI instead of ΔI/I and how they can compare signal intensities in this way. Moreover, signal amplitudes are compared although different frequencies are being used. EDMR shows a weak MW frequency dependence and this should then be considered (see e.g. Fukui et al. Volume 149, Issue 1, March 2001, Pages 13-21).*

We understand that the relative current change is reported for comparison in varying experimental conditions. However, since all experiments in this manuscript have been performed with exactly the same sample and exactly the same electrical settings for the CP, there would not be a difference in plotting ΔI or ΔI/I, as the I is essentially unchanged. The "arbitrary units" used in the manuscript are the same for all EDMR intensities presented here, as we have now specified in the experimental section. Hence, since we give the units in arbitrary units, this does not matter.

We have added the following on line 166-169:

*Thus, all experiments are performed with the same CP settings and thus delivering the same CP current to the TIA and lock-in amplifier so that all experiments can be directly compared. Note that EDMR intensities are provided in arbitrary units, but that these are the same for all experiments. Typical changes in CP current upon magnetic resonance are of the order of $5 \times 10^{-4}$ in these devices.*

The frequency-dependence of EDMR is expected to be small from 9.85 to 14 GHz, as this is reported in the provided manuscript to level off already around 1 GHz. We have added a remark about this on lines 283-285 of the revised manuscript

*Note that the frequency dependence of EDMR is expected to be small from 9.85 GHz to 14 GHz, as this has been demonstrated to level off already around 1 GHz.(Fukui et al., 2001)*

6. *The origin of the EDMR signal is not discussed. What type of a defect is observed and how does this defect inter act with the CP condition. The authors have to describe details about the mechanism (is it a spin-pair mechanism (KSM)?) or cite accordingly. If this is not a KSM process, the magnetic field dependence will be strong and must be considered.*

We discussed this briefly in section 2.1.4: charge pumping and EDMR detection. Although the exact mechanism of CP-EDMR is not yet known in detail, despite many publications that can be found in literature (see introduction where we cite them), the CP-EDMR is usually indeed interpreted in view of the KSM model and spin-pair formation before recombination. We have added the following statement in our manuscript regarding this on line 171:

*The recombination current detected in the CP experiment depends on the spin state of the carrier pairs involved in recombination processes in the transistor channel region **according to the Kaplan-Solomon-Mott model.**( Kaplan et al., 1978; Boehme and Malissa, 2017) The spins can be flipped by absorption of MW quanta if the condition of magnetic resonance is fulfilled, i.e. when $h\nu = g\mu_B B_0$ (with h for Planck's constant, ν the MW frequency, g the Landé g-factor, $\mu_B$ the Bohr magneton, and $B_0$ the applied static magnetic field).*

This manuscript is not intended to unravel the mechanism behind CP-EDMR, but to compare between the state-of-the-art cavity-based and EDMRoC measurements of this specific spin dependent current, keeping all CP settings the same. The mechanism of the CP is not essential for the understanding and is therefore not treated any further in the current manuscript.

7. *Since this paper deals with spectroscopy, the spectroscopically determined line parameters should be stated in this paper and compared to literature values.*

We had already provided the line widths of the experiment, currently on line 296.

8. *A large portion of the discussion is about the low-cost approach. How does the EDMRoC compare to EDMR performed with a strip-line approach as is frequently used? This setup is extremely easy to established and does not require a by far more complex device like the EPRoC. The authors should discuss this to some extend.*

As discussed above in the reply to reviewer 1, the main difference is that the MW frequency is generated by the 32nd harmonic of a reference frequency, therefore only needing an approximately 430MHz reference frequency to then obtain the higher frequency 13-14GHz microwaves. Our present use of the SMC 100B RF generator is an overkill. Previous work already demonstrated that such a reference frequency could be delivered by a single PCB at very low cost. Combining it with all the necessary power supplies and lock-in on another PCB, one can fit the whole spectrometer into 10x10x10 $cm^3$ as demonstrated previously. [doi: 10.1109/ISSCC.2016.7418114] We have highlighted this further on page 395-399 in the manuscript.

**Textual comments:**

> *Line 28: The statement "…but also more complex devices…" is too bold. A solar cell, for instance, can easily exceed the complexity of a transistor, because this device has to be considered under illumination (large quasi Fermi splitting) and under high charge injection. Leave out the word "more complex"*

We have made this change.

> *Line 45: The authors should report on previous works where the two suggested ways to improve the sensitivity are described.*

Our statement is not about higher sensitivity, but higher selectivity compared to other EDMR schemes (diode current or BAE) for MOSFETs using the multidimensional parameter space of CP and the extensive knowledge built up in the analysis of CP experiments. There is up to now only limited data available that demonstrates this selectivity, hence the word 'potentially' in the sentence. However, some correlations have already been shown as reported in the references at the end of the previous sentence: (Aichinger et al., 2013; Anders et al., 2020; Bittel et al., 2011; Cochrane et al., 2013; Gruber et al., 2014, 2018). More work is going on to distinguish between different traps responsible for the EDMR spectra in these devices.

> *Line 53: How compact and expensive is the setup? The word "extreme" demands an explanation. And what other setup do the authors compare their's to? For example, the antenna used by McCrory for the EDMR applications is rather small and not to costly. Please comment.*

As mentioned above, the EDMRoC and a simple PCB for the reference frequency could be delivered at costs of a few hundred euros, to less than hundred euros if a larger amount of units is ordered. Combining it with all the necessary power supplies and lock-in on another PCB, one can fit the whole spectrometer into 10x10x10 $cm^3$ as demonstrated previously. [doi: 10.1109/ISSCC.2016.7418114] We have highlighted this further on page 395-398 in the manuscript

> *Line 60: Which Si technology was used for the fabrication of the chip?*

The chip was manufactured using a commercial 130nm RF-CMOS technology from GlobalFoundries. We have specified this now in the text on lines 83-84

*Line 74: Omit the word quantitative. In this report the argumentation is mainly qualitative.*

We have changed the wording in this sentence.

*Line 82: What type of TIA was used?*

The TIA description was already provided on line 189, it is a SR570 system, very often applied in EDMR research. We have now also added it at this point (currently line 85)

*Line 83: What is an application specific ASIC?*

As spelled out at first use in our manuscript (line 87 in current version) ASIC is short for 'application-specific integrated circuit', hence this is just the full writing of the acronym. It is a main board containing all the electronics for the EPR application and is general terminology in this field of electronic engineering.

*Line 86: Is there any reference to the custom made PLL? Can the authors give details?*

We added the following explanation to line 91-94:

*The PLL uses a commercial phase-frequency detector (HMC439 – Analog Devices) to compare the divided phase of the on-chip VCO with the phase of a reference source. The output of HMC439 is low-pass filtered using an operational amplifier (THS4304 – Texas Instruments) in combination with passive R-C circuits. Overall, the PLL has a bandwidth of 1 MHz, which is sufficient for CW EPR experiments.*

*Line 108: I suppose you mean a sweepable magnet?*

Yes, we use 'variable' and 'sweepable' as synonyms, but for simplicity we have now replaced it by sweepable.

*Line 145: Should read Fig. 2e*

Corrected.

*Line 173: The sentence is incomprehensible. Please rephrase.*

We assume that the incriminated sentence is:

"The final steps in the signal generation and treatment are common to both approaches, allowing as much as possible a direct comparison of signal strengths."

This sentence was meant to emphasize that, for best comparison, all following instruments and their settings are kept identical between cavity-based EDMR and EDMRoC, namely amplification and low-pass filtering using the TIA, phase-sensitive detection by the lock-in and registration via the SPU unit of the EPR spectrometer. (As was detailed in the next sentence.)

In the new submission this is reformulated on lines 187-188:

*In order to reach a meaningful comparison between the signal intensities from the cavity-based EDMR and from the EDMRoC experiments, identical signal treatment and collection is applied in all measurements, as follows:*

*Line 177: Please provide the settings and details of the TIA (see comment line 82).*

The specific type of TIA used in our measurements was already spelled out in subsection 2.1.5, line 174 (in the initial manuscript). Care has been taken to use the same settings for all experiments in the comparison between cavity-based EDMR and EDMRoC, as listed on line 194-195:

*For the settings of the TIA we use the low-noise gain mode with a 12 dB low pass filter of 3kHz and a sensitivity of 2µA/V.*

> **Line 208-216: What is the distance between EPRoC and the MOSFET? Has this been considered for the calculation? The MOSFET only covers a part of the coils of the resonator array. How does this affect the results?**

Indeed, the MW field on the MOSFET structure strongly decreases with distance above the MW emitting chip, as already stated in subsection 2.1.1 (line 90 of initial submission) and discussed in more detail in subsection 2.2.1 (lines 203-205 initial submission) on the basis of a previous characterization of the MW chip. While the lateral positioning is sufficiently repeatable using the alignment hole and pin (see Figures 2(a) and 2(d)), our mounting procedure leaves an uncertainty in the resulting distance between the two devices, as was revealed by the different saturation curves obtained after independent mounting (discussed in lines 227-230, initial submission, and Appendix F). The distance dependence of the transition probabilities is also modelled in Appendix H, Figure H2, for the static magnetic field either parallel to or perpendicular to the sample surface. The experimental ratio between the two intensities (Appendix H, Figure H1) is in good agreement with a distance of at least 15 µm, as discussed in Sec. 3 (lines 341-346, initial submission).

As also discussed in Section 3 (lines 333-334, initial submission), the sample mounting calls for improvement (spacers or accurate mechanics) to gain proper control on the distance between both devices.

The width of the devices fits well with that of the VCO array and the lateral positioning of the sample relative to chip is sufficiently repeatable. However, the longs dimensions are not completely matching as is clearly shown in Figure D1. This does not affect the experiment as the MW field is reaching essentially the whole MOSFET area. It eventually would admit for larger samples for effective MW excitation. It should be noted that the MW emitting device is scalable by using different number of coils adapted to specific applications.

> **Fig. 3b,c,d: The y-axis should read DI/I or DI.**

As mentioned above, as all EDMR experiments are performed with the same electrical settings, with repeatable value of the CP current I, DI/I or DI are equivalent albeit arbitrary units. The units on the axis are linearly proportional to DI and significant for comparison of the EDMR intensity in the different spectra in this article.

> **Line 241: What was the modulation frequency? Were any phase shifts observed? In particular, the EDMR signal often also has a non-resonant signal which would only appear in the BM signal. Is such a signal observed?**

The modulation frequency is mentioned in the manuscript on line 175 – it was always kept at 730 Hz in either BM or FM. We always measured the in-phase and out-of-phase signals from the LIA simultaneously and, as already stated in line 187 (revised manuscript), no significant phase shifts were found.

The other question was already discussed on lines 298-304 of the manuscript.

> **Line 260-265: As stated above, this discussion is critical with respect to the definition of the signal intensity. Please comment.**

This was already discussed as mentioned above. The background was subtracted in both cases to compare the signal intensity. We realized we did not mention this in the original manuscript, so we now added this in the caption of Figure 3, 4, and 5.

> *Line 278: I assume that the authors refer to g-strain, that leads to the line broadening. This effect can be easily simulated. With that, further details of the line parameters should be given and compared. What do the authors mean by "leading to reduced resolution of sidebands"? From Fig. 4c it becomes obvious that no field dependent process determines the line shape. How does this correlate with "sidebands"?*

The weak side-bands originate from the hyperfine interaction sometimes ascribed to nearby Si nuclei, which can be observed at X-band frequency but not at the higher 13GHz frequency due to the enhanced *g*-strain broadening. This is general in frequency-dependent EPR and EDMR experiments. We have changed the sentence on lines 294-297 to:

*It is worth reminding that the resonance occurs at higher magnetic fields in EDMRoC due to the higher MW frequency (13.28GHz) compared to EDMR in the X-band high-Q resonator (9.85GHz). Also, the effects of **g-**strain inhomogeneity appear in the linewidth of the resonance (18 MHz in EDMRoC vs. 12 MHz in resonator EDMR), leading to reduced resolution of **the hyperfine** sidebands as visible in Fig. 3(c) and Fig. 3(d).*

> *Line 312: The authors should fit the lines of all three measurements and then give quantitative numbers on the field inhomogeneity. This adds to the quality of the paper and makes it more quantitative.*

The line width in the permanent magnet assembly is slightly larger (18.9 MHz) than in the electromagnet (16.7 MHz) as we had mentioned in the manuscript. We have now added these line widths to the text on line 334:

*...permanent magnet assembly (18.9 MHz versus 16.7 MHz in the electromagnet) most likely*

> *Line 327: In the discussion the mechanism of the observed EDMR lines should be mentioned or cited, if well accepted.*

Different interpretations can be found in the SiC CP-EDMR literature on the origin of the observed lines and moreover also different lines are observed in different devices. In these circumstances it is not straightforward, and also irrelevant for the current manuscript, to delve into the mechanisms of the EDMR lines.

> *Line 343: How would these parallel components of B1 change the saturation behavior of EDMR and why are they not considered?*

The power calibration experiment based on saturation was performed for B0 parallel to the surface with transition probability $P_y \sim B_{1,x}^2 + B_{1,z}^2$ (see discussion in Section 2.2.1, lines 217-230 of initial manuscript; also Appendices B, F and H). As the components of the MW fields from the MW chip, in any point in space above the MW chip, are proportional to the current in the coils, the relative contribution of each of the $B_1$ components is not changing, and the saturation behavior is a combined effect of the $B_1$ components that contribute to the transitions probabilities. As we calibrated the power dependence in EDMRoC relative to that in cavity-based EDMR, leading to an effective mutual scaling, the derived power dependence for EDMRoC is a combined effect of the $B_1$ components.

The parallel components are properly considered in the $B_1$ fields simulations (Appendix G, Figure G1), and in calculating the distance dependence of the transition probabilities $P_y$ and $P_z$ for the two field orientations from these field simulations (Appendix H, Figure H2).

**Line 423: Please label the y-axis and the two resonance curves. Is the solid line a fit?**

Yes, this is a Lorentzian fit, we adapted the caption of the figure. Since the picture shows a normalized and inverted curve of the reflected MW intensity from the cavity (the so-called "dip" of the cavity), the y-axis label cannot be labelled. Note that only the linewidth is needed to determine the Q-factor of the cavity. We have added this description to the caption of the figure.

**Fig. F1: The distance of Chip to sample is very critical. How do the authors take this experimental uncertainty into account? Please comment.**

As we mention in the manuscript on lines 334-336 and as we demonstrate with this figure, we cannot take it into account yet. However, it is possible to implement better mechanical mounting to control the chip-to-sample distance.

**Fig. H1: What leads to the shift and narrowing of the line? Please discuss in the text.**

The shift reveals a small $g$-anisotropy, $\Delta g \cong 0.00025$, of the detected defect, which can be resolved even at X-band frequency. There is also a difference in linewidth, which may be ascribed to changes in unresolved nuclear hyperfine interactions. It is thus not inherent to the design of the chip. The data presented in this appendix specifically aims at the demonstration of a non-negligible contribution of $B_1$ components parallel to the sample surface.

We have clarified this in the caption of Figure H1.